# MIN-MAX ZERO-SHOT MULTI-LABEL CLASSIFICATION

## ABSTRACT

In many classification problems, acquiring labeled examples for many classes is difficult, resulting in high interest in zero-shot learning frameworks. Zero-shot learning (ZSL) is a problem setup where at test time, a learner observes samples from classes that were not observed/trained in the training phase and is required to predict the category they belong to. Zero-shot learning transfers knowledge from seen classes (observed classes in the training phase) to unseen classes (unobserved classes in the training phase but present in the testing phase), reducing the human labor of data annotation to build new classifiers. However, most zero-shot learning researches target single-label classification (multi-class setting). There are a few studies on multi-label zero-shot learning due to the difficulty in modeling complex semantics conveyed by a set of labels. We propose a novel probabilistic model that incorporates more general feature representation (e.g., Word-Net hierarchy, word2vec features, convolutional neural network features (layer-wise), and co-occurrence statistics) and learns the knowledge transfer in terms of data structure and relations. We also investigate the effect of leveraging different CNN layers' features. Our experimental results prove the efficacy of our model in handling unseen labels. We run additional experiments to analyze the flat-sharp minima convergence of methods as a generalization factor. Our study suggests that our proposed method converges to flat minima resulting in strong generalization.

## 1 INTRODUCTION

Supervised learning methods require a large number of labeled instances for training purposes inducing a high annotation cost. Developing classification algorithms that reduce the annotation cost has recently attracted more attention. In Zero-shot learning (ZSL), the classes that the classifier observes and is trained on are called *seen* classes. Conversely, the classes that the classifier does not observe in the training phase and just tests on them in the testing phase are called *unseen* classes. This ability to classify instances of an unseen class leveraging the knowledge from seen classes is called zero-shot learning Socher et al. (2013).

The attribute description of the class labels is a primary source in ZSL to bridge the gap between seen and unseen classes Lampert et al. (2009; 2014); Fu et al. (2015); Romera-Paredes & Torr (2015). The attributes can carry transferable information across classes since they are defined based on common and specific characteristics of different category concepts. Hence, many ZSL methods employ different attribute types and demonstrate impressive results. For instance, textual descriptions from Wikipedia articles Qiao et al. (2016); Akata et al. (2016), word embedding vectors trained from large text corpus using natural language processing (NLP) techniques Akata et al. (2015b); Frome et al. (2013); Xian et al. (2016); Zhang & Saligrama (2015); Al-Halah et al. (2016), co-occurrence statistics of hit-counts from search engine Rohrbach et al. (2011); Mensink et al. (2014) and WordNet hierarchy information of the labels Rohrbach et al. (2010; 2011); Li et al. (2015) have been used. However, majority of these studies investigate multi-class ZSL, while the more challenging multi-label ZSL problem has received less attention Mensink et al. (2014); Zhang et al. (2016); Lee et al. (2017).

Multi-Label (ML) classification is applied in diverse domains like recommender systems Prabhu & Varma (2014) and computer vision Wang et al. (2016). Most existing multi-label methods consider structural relations on binary label outputs to leverage the label correlation in domains with a large number of labels Yu et al. (2014); Rai et al. (2015). Multi-label classification is more challenging when all the labels are not available at training time Zhang et al. (2016); Mensink et al. (2014). In many real-world applications (e.g., recommender systems and computer vision), the label set keeps increasing, and re-training the classifier might be time-consuming and expensive. Gaure proposed a

probabilistic framework Gaure et al. (2017) leveraging the co-occurrence statistics of the seen labels with both seen and unseen labels. It is an EM-based algorithm in which every parameter update requires solving a weighted ridge-regression problem.

The relationship/structure between seen and unseen classes is ignored in most multi-label ZSL methods. For instance, a seen and unseen class can share similar co-occurrence and taxonomy pattern resulting in classifying the unseen class correctly in the test phase. We propose a novel framework that relies on the data relationship among different classes (structured prediction). To account the pattern and structure among seen and unseen classes, we propose to model the classes (graph nodes) and their relationship (graph edges) as a complete graph. Transferring the relationships among classes is one of the main novelties of our method.

We conduct extensive experiments to investigate the performance of our proposed approach. The empirical results demonstrate the effectiveness of our proposed approach, considering different baselines and datasets. Finally, We run convergence analysis based on the flat/sharp minima aspect and demonstrate our proposed method's convergence to flat minima. It presents the importance of convergence to flat minima for strong generalization support.

## 2 BACKGROUND AND RELATED WORK

In this section, we explain ZSL and the motivation behind our approach. We explain multi-label classification and related works regarding zero-shot learning in Appendix for the sake of space. **Zero-Shot Learning (ZSL):** In supervised classification, sufficient labeled samples for target classes are required in the training phase Bishop (2006). Collecting and labeling a large number of samples is expensive in many classification tasks. For example, it is hard/dangerous to take images of samples like wild animals in the oceans, plants in the high depth of oceans, or the high height of mountains. Moreover, many objects "in the wild" do not occur frequently enough to collect and label a large set of representative samples to build the corresponding classifier Changpinyo et al. (2016). Zero-shot learning (ZSL) addresses this problem by transferring knowledge from abundantly labeled source classes to help build classifiers for target classes with no available labeled samples. In more detail, there are no labeled data for some classes in the training phase, and the learner is asked to predict those classes at test time. The primary assumption is that there should be some similarity between the unseen and seen classes. Although they are different classes but related via some features and auxiliary information.

ZSL builds a classifier to predict the presence or absence of unseen classes for test data. The critical factor in ZSL is effectively transferring knowledge between seen and unseen classes. Our approach employs the Min-Max formulation Asif et al. (2015) and models the multi-label ZSL as learning the structure and the relationships among the nodes in a complete graph. In addition, we employ auxiliary resources like word2vec, WordNet hierarchy, and features from different convolutional neural network layers in the model feature representation. However, the solid dependency of the model on these resources is controlled by modeling the problem as a min-max zero-sum game Asif et al. (2015); Behpour et al. (2018).

## 3 MIN-MAX MULTI-LABEL ZSL

We propose a novel approach for multi-label zero-shot learning based on leveraging robust cuts Blum & Chawla (2001); Behpour et al. (2018). A labeled training dataset (seen classes) and a test dataset with a set of seen and unseen classes are given in this setting. The objective is to learn a multi-label classifier from the training set (with seen classes) and generalize it to the test set with unseen labels.

This generalization requires an intermediate feature representation shared between training and test classes. Hence, a model that can define the importance of features for the classification task is required. We propose a robust ZSL classifier capturing the most effective and implicit feature similarities leveraging graph representation to make better generalizations and predictions on unseen classes. Our approach builds on the min-max robust learning, which approximates the training data labels with a worst-case distribution while still resembling training data properties Topsøe (1979); Grünwald & Dawid (2004); Asif et al. (2015). The learning task in multi-label classification is modeled as a min-max zero-sum game between a maximizer and a minimizer based on minimum cost graph cuts. Furthermore, it deploys a joint representation of the features in its feature representation

to embed the label co-occurrence statistics in the training phase. This graph representation and the min-max training process provide informative knowledge transfer in the ZSL framework.

### 3.1 Problem Definition and Notations

We first describe how a multi-label (ML) classification is modeled in our method following Asif et al. (2015). Next, we present our proposed ML-ZSL framework and then explain our feature representation that plays a significant role in our method's outstanding performance and accuracy.

The ML annotation prediction is modeled as a Markov network with nodes indicating the labels/classes, edges representing the pairwise relationship among the nodes, and two extra nodes: "0" and "1". These two extra nodes are connected to all other nodes in the graph. In this framework, the parameters are learned by the classifier to make a min-cut in the graph resulting in present nodes (connected to node "1") and absent ones (connected to node "0"). For better illustration, please consider a dataset with six labels: $\{Lion, Sheep, Panda, Cow, Dog, Dolphin\}$ as our running example. We present an overview of the Markov network for our running example in figure 1.

As it is shown in figure 1, a Markov networks with $n$ nodes corresponding to the $n$ predicted variables, $\mathbf{y} = (y^1, \ldots, y^n)$, chosen from a fixed set of labels $y^i \in \mathcal{Y}, \forall i \in [n]$, where $[n] = \{1, \ldots, n\}$. In our notation, we denote the corresponding *random variables* for these label variables by $\mathbf{Y} = (Y^1, \ldots, Y^n)$, and present vectors and multivariate variables in bold. Given information variables are demonstrated using a single vector, $\mathbf{x} \in \mathcal{X}$, with a corresponding random variable denoted as $\mathbf{X}$.

In this setting, the task is to make predictions for unseen (presented as "u") classes/labels $\mathbf{y}^{\mathbf{u} \notin \{\mathbf{1}, \ldots, \mathbf{n}\}}$ given an input $\mathbf{x}$ and a set of $m$ training example pairs, $(\mathbf{x}_j, \mathbf{y}_j)_{j \in [m]}$. As indicated, there is an associated label vector $y$ for a training sample $x$ corresponding to different classes of the dataset. For instance, Figure 2 demonstrates the label vector $y$ where labels *sheep*, *cow* and *dog* are present in the image. Here $I()$ is an indicator function whose value is 1 only if the an inner logical expression is true.

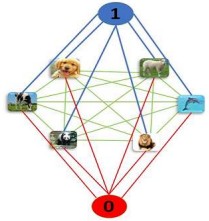

$$\mathbf{y} = \begin{bmatrix} I(y_{lion} = 1) \\ I(y_{sheep} = 1) \\ I(y_{panda} = 1) \\ I(y_{cow} = 1) \\ I(y_{dog} = 1) \\ I(y_{dolphin} = 1) \end{bmatrix} = \begin{bmatrix} 0 \\ 1 \\ 0 \\ 1 \\ 1 \\ 0 \end{bmatrix}$$

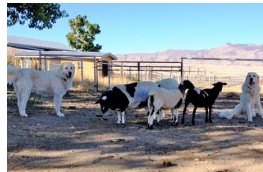

**Figure 2:** An image including sheep, cow, and a dog is represented as label vector y.

**Figure 1:** Markov Network model for Multi-Label prediction.

### 3.2 Feature Representation

Feature representation plays a key role in finding an intermediate clue for generalization from seen classes to unseen ones. A new/unseen class can be demonstrated by matching the similarities of seen classes. Thus, incorporating semantics into the classification process is crucial to achieving high prediction accuracy. Four feature representations (joint, layer-wise, pairwise, and hierarchy) are employed in our approach to learning the features representation and the features' relationship between different classes. Given a training sample $x$, we present its feature representation by $\phi_x$, which is employed to define our other features. For instance, the feature vector provided by Mulan dataset Tsoumakas et al. (2011) for figure 2 can be counted as feature vector $\phi$ for this image.

We define joint features following Kim et al. (2012) based on the presence of the class instance in the sample as equation 1. In our equations, $I()$ is an indicator function whose value is 1

$$\phi_{joint} = \begin{bmatrix} I(y^{lion} == 1)[\phi_x] \\ I(y^{sheep} == 1)[\phi_x] \\ \ldots \\ I(y^{dolphin} == 1)[\phi_x] \end{bmatrix}. \quad (1) \qquad \phi_{layer-wise} = \begin{bmatrix} I(y^{lion} == 1)CNN_K(x) \\ I(y^{sheep} == 1)CNN_K(x) \\ \ldots \\ I(y^{dolphin} == 1)CNN_K(x) \end{bmatrix}. \quad (2)$$

only if the inner logical expression is true. The layer-wise features represent the features of layer $k$ from a convolutional neural network represented as $CNN_K(x)$ for the given image $x$.

Recently, convolutional neural networks (CNN) have shown outstanding performance in providing informative features for images. Every layer of CNN extracts different features. The lower layers represent the structure/texture of the image input, and the upper layers provide a general/high-level representation. Existing ZSL frameworks that employ CNN features focus only on features from fully connected layers and ignore the significance of the lower convolutional layers. One of the novelties of our method is that we extract the features from different layers and show that employing features from other layers of the convolutional layers in addition to the top layers results in better performance

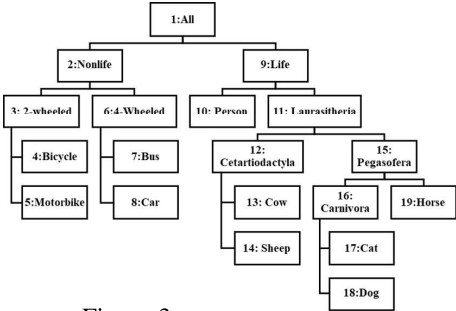

Figure 3: The VOC2006 taxonomy.

and accuracy in ML-ZSL. Different classes share the same or similar structures, specifically when they are from the same ancestors. The CNN lower and top convolutional layers capture these structures and can be generalized well to unseen classes. In our experiment, we consider different CNN layers' features and demonstrate the outstanding performance of these feature representations.

We present the pairwise (edge) features by leveraging word2vec [1] features to present the relationship between two classes/ nodes $i$ and $j$. The pairwise feature representation based on word2vec features is computed as follows:

$$\phi_{Wi,j}(y^i, y^j) = I(y^i \neq y^j)|\mathbf{word2vec}(i) - \mathbf{word2vec}(j)|^{-1}, \tag{3}$$

Here, $\phi_W$ refers to pair-wise features based on word2vec feature representations for the edge between nodes $i$ and $j$.

Our last type of features, taxonomy features, detail the inner-class similarities and dependencies that improve classification performance. We deploy taxonomy features in addition to joint, layer-wise, and pairwise features to provide a comprehensive feature representation for the classifier. In this setting, we are given a taxonomy $T$ demonstrated by an arbitrary directed graph $(V_t, E_t)$ where $V_t = (v_1, v_2, ..., v_{|T|})$ represents the graph nodes set and $E_t$ presents the graph edges set. The set of nodes from the root node to a leave node $y$ is presented as $\pi(y)$. We employ $T(y)$ to represent $\pi(y)$ structure as follows: $T_j(y) = \begin{cases} 1 & v_j \in \pi(y) \\ 0 & \text{otherwise} \end{cases}$.

For instance, the taxonomy vector of figure 3 is presented as T=[All, Nonlife, 2-wheeled, Bcycle, Motorbike, 4-wheeled, Bus, Car, Life, Person, Laurasitheria, Cetartiodactyla, Cow, Sheep, Pegasofera, Carnivora, Cat, Dog, Horse]. Following this taxonomy vector, class *sheep* is presented as T(sheep) = [1, 0, 0, 0, 0, 0, 0, 0, 1, 0, 1, 1, 0, 1, 0, 0, 0,0, 0]'. In T(sheep), the nodes " All, Life, Laurasitheria, Cetartiodactyla, and sheep " presented in the path from "root" to "sheep" get value "1" and the other nodes get value "0".

We subsume the structural taxonomy information and encode the taxonomy paths by leveraging a joint feature representation following Kim et al. (2012) that is given by tensor product class features and class indicator $[[v_i \in \pi(y)]]$ as follows:

$$\psi(x, y) = \phi(x) \otimes T(y) = \begin{bmatrix} \phi(x)[v_1 \in \pi(y)] \\ \phi(x)[v_2 \in \pi(y)] \\ ... \\ \phi(x)[v_{|V|} \in \pi(y)] \end{bmatrix}. \tag{4}$$

### 3.3 MIN-MAX METHOD FORMULATION

Min-max prediction methods Grünwald & Dawid (2004) leverage two players, maximizer and minimizer, and demonstrate a maximizer distribution approximation of the training data labels/classes, $P_{maxi}(y_{maxi}|x)$, while seek a minimizer distribution $P_{mini}(y_{mini}|x)$.

The minimizer distribution over possible labeling of the samples minimizes the expected loss against the worst-case distribution chosen by the maximizer distribution (over possible labeling of the

---

[1]https://code.google.com/p/word2vec/

samples):

$$\min_{P_{mini}} \max_{P_{maxi}} \mathbb{E}_{x \sim P_{Data}; y_{maxi}|x \sim P_{maxi}; y_{mini}|x \sim P_{mini}} \left[ \text{loss}(Y_{mini}, Y_{maxi}) \right] \text{ such that: } \mathbb{E}_{x \sim P_{Data}; y_{maxi}|x \sim P_{maxi}} \left[ \phi(X, Y_{maxi}) \right] = \tilde{\mathbf{c}}.$$

Here the label vector prediction by maximizer and minimizer players are shown as $y_{maxi}$ and $y_{mini}$ respectively. $\phi$ is the feature representation explained in detail in section Feature Representation. The maximizer player, $y_{maxi}$, is constrained by empirical data statistics, $\tilde{\mathbf{c}}$— using equality or inequality constraints. $\tilde{c}$ is the empirical data statistics based on the feature function of training samples, and it is computed as follows:

$$\tilde{c} = \mathbb{E}_{\mathbf{X}, Y \sim P_{Data}} \left[ \phi(\mathbf{X}, Y) \right] = \frac{1}{m} \sum_{i=1}^{m} \phi(\mathbf{x}_i, y_i). \tag{5}$$

$P_{Data}$ presents the empirical distribution of $\mathbf{X}$ [and $\mathbf{Y}$] in the training data set. This Min-Max formulation provides better performance in practice for classification Asif et al. (2015) due to aligning the training objective to the loss measure instead of surrogate losses like hinge loss. It is one of the important min-max method features that qualifies it for zero-shot learning. The knowledge transfer in ZSL gets more accurate when the learning and optimization of seen classes are done based on the **exact-loss** optimization method (not loss-approximation methods like SVM and logistic regression).

Another strength point of this method lies in providing a probability distribution over classes ($y^k$ presents the $k$th label/class in label vector $y$, $1 \le k \le n$). This probability distribution is beneficial in the ZSL setting as it represents the correlations between known and unknown label variables, $P(y^i, y^j)$ where $y^i \in$ seen classes, and $y^j \in$ unseen classes. It plays a key role in generalizing seen classes to unseen classes in the ZSL setting. Other probabilistic methods like conditional random fields Lafferty et al. (2001) are not computationally effective compared to the Min-Max method. In the remainder of this section, we present the Min-Max method for the ZSL setting.

We present the structure of our method in Figure 4. As it is shown in Figure 4, the classifier is trained to make the right min-cut on training samples graphs (including only seen classes) in the training phase, and it predicts min-cut on testing samples graphs (including only unseen classes).

The number of nodes (classes) in the testing phase can differ from the training phase. When the test set includes both seen and unseen classes, it is called *generalized* ZSL.

Our Min-Max method provides strong performance in making the right min-cut in the modeled graph as it leverages exact-loss optimization, the structure of the data in modeling the problem, and appropriate feature representation. We present Min-Max ZSL in the next section in full detail.

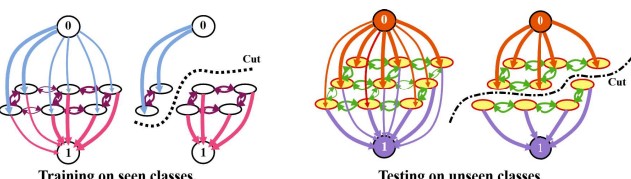

**Training on seen classes.**  **Testing on unseen classes.**

**Figure 4:** The structure of the Min-Max ML-ZSL applying graph cut to transfer the knowledge. The graph nodes are classes/labels, which are two disjoint sets. The edge weights are identified with feature potentials.

## 3.4 Min-Max Multi-Label Zero-Shot Learning

We model the ML-ZSL problem as a binary-valued Markov network with non-negative pairwise potentials. Two additional nodes, source (0) and target (1), are added to the network to indicate present and absent nodes/classes in every sample. Therefore, all network nodes (class nodes) are connected to these two nodes. Edges from the source and to the sink nodes are weighted based on unary (joint, layer-wise, and taxonomy) features potentials. The edges between predicted variables are weighted based on the pairwise features potentials. Minimum-cut/maximum-flow algorithm can provide the maximum value assignment of the network by dividing the nodes into two mutually exclusive sets connected to the nodes source or target (e.g., figure 4).

We consider Hamming loss as the loss function in this study and experiments. Hamming loss computes the fraction of the correct predicted labels in the multivariate annotation task.

$$\text{loss}(\mathbf{y}_{mini}, \mathbf{y}_{maxi}) = \frac{1}{n} \sum_{i} I(y_{mini}^i \ne y_{maxi}^i). \tag{6}$$

Our features are presented using $\phi$ including $\phi_{taxonomy}(\phi_t)$, $\phi_{joint}$, $\phi_{layerwise}(\phi_{lw})$, and $\phi_{pairwise}(\phi_{pw})$ in our model formulation. Min-Max ML-ZSL game formulation is presented as:

$$\min_{P_{mini}(\mathbf{Y}_{mini}|\mathbf{X})} \max_{P_{maxi}(\mathbf{Y}_{maxi}|\mathbf{X})} \mathbb{E}_{\substack{\mathbf{X},\mathbf{Y} \sim P_{Data}, \\ \mathbf{Y_{mini}}|\mathbf{X} \sim \mathbf{P}_{mini}, \\ \mathbf{Y_{maxi}}|\mathbf{X} \sim \mathbf{P}_{maxi}}} [\text{loss}(\mathbf{Y}_{mini}, \mathbf{Y}_{maxi})]$$

such that: $\mathbb{E}_{\mathbf{X} \sim P_{Data}; \mathbf{Y_{maxi}}|\mathbf{X} \sim \mathbf{P}_{maxi}} \left[ \phi_t^i(Y_{maxi}^i, \mathbf{X}) \right] = \mathbb{E}_{\mathbf{X}, \mathbf{Y} \sim P_{Data}} \left[ \phi_t^i(Y^i, \mathbf{X}) \right], \forall i \in [n]; t\text{:}taxonomy$

$\mathbb{E}_{\mathbf{X} \sim P_{Data}; \mathbf{Y_{maxi}}|\mathbf{X} \sim \mathbf{P}_{maxi}} \left[ \phi_{joint}^i(Y_{maxi}^i, \mathbf{X}) \right] = \mathbb{E}_{\mathbf{X}, \mathbf{Y} \sim P_{Data}} \left[ \phi_{joint}^i(Y^i, \mathbf{X}) \right], \forall i \in [n];$

$\mathbb{E}_{\mathbf{X} \sim P_{Data}; \mathbf{Y_{maxi}}|\mathbf{X} \sim \mathbf{P}_{maxi}} \left[ \phi_{lw}^i(Y_{maxi}^i, \mathbf{X}) \right] = \mathbb{E}_{\mathbf{X}, \mathbf{Y} \sim P_{Data}} \left[ \phi_{lw}^i(Y^i, \mathbf{X}) \right], \forall i \in [n]; lw\text{:}layer\text{-}wise$

$\mathbb{E}_{\mathbf{X} \sim P_{Data}; \mathbf{Y}_{maxi}|\mathbf{X} \sim \mathbf{P}_{maxi}} \left[ \phi_{pw}^{i,j}(Y_{maxi}^i, Y_{maxi}^j, \mathbf{X}) \right] \leq \mathbb{E}_{\mathbf{X}, \mathbf{Y} \sim P_{Data}} \left[ \phi_{pw}^{i,j}(Y^i, Y^j, \mathbf{X}) \right], \forall i, j \in [n]; pw\text{:}pairwise.$  (7)

In formulation 7, the maximizer aims to make the most uncertain label distribution resulting in a large amount of expected loss for the maximizer. However, leveraging the constraints defined on features and feature relations put the maximizer in a predictable direction.

These constraints can be incorporated as lagrangian potentials with learning parameters $\boldsymbol{\theta}$ including $\theta_{taxonomy}(\theta_t)$, $\theta_{joint}$, $\theta_{layer-wise}(\theta_{lw})$, and $\theta_{pairwise}(\theta_{pw})$ by deploying strong duality Boyd & Vandenberghe (2004) and the method of lagrangian multipliers:

$$\min_{\{\boldsymbol{\theta}_i\}, \{\boldsymbol{\theta}^{i,j}\} \leq \mathbf{0}} \min_{P_{mini}(\mathbf{y}_{mini}|\mathbf{x})} \max_{P_{maxi}(\mathbf{y}_{maxi}|\mathbf{x})} \mathbb{E}_{\substack{\mathbf{X},\mathbf{Y} \sim P_{Data}, \\ \mathbf{Y_{mini}}|\mathbf{X} \sim \mathbf{P}_{mini}, \\ \mathbf{Y_{maxi}}|\mathbf{X} \sim \mathbf{P}_{maxi}}} \left[ \text{loss}(\mathbf{Y}_{mini}, \mathbf{Y}_{maxi}) + \sum_i \boldsymbol{\theta}_t^i \cdot (\boldsymbol{\phi}_t^i(Y_{maxi}^i, \mathbf{X}) - \boldsymbol{\phi}_t^i(Y^i, \mathbf{X})) \right.$$

$$+ \sum_i \boldsymbol{\theta}_{joint}^i \cdot (\boldsymbol{\phi}_{joint}^i(Y_{maxi}^i, \mathbf{X}) - \boldsymbol{\phi}_{joint}^i(Y^i, \mathbf{X})) + \sum_i \boldsymbol{\theta}_{lw}^i \cdot (\boldsymbol{\phi}_{lw}^i(Y_{maxi}^i, \mathbf{X}) - \boldsymbol{\phi}_{lw}^i(Y^i, \mathbf{X}))$$

$$+ \sum_{i \neq j} \boldsymbol{\theta}_{pw}^{i,j} \cdot (\boldsymbol{\phi}_{pw}^{i,j}(Y_{maxi}^i, Y_{maxi}^j, \mathbf{X}) - \boldsymbol{\phi}_{pw}^{i,j}(Y^i, Y^j, \mathbf{X})) \Big].$$  (8)

For a detailed proof on transforming Eq. 7 to Eq. 8, please see appendix, section "Eq. 2 to Eq. 3 transformation proof".

Constructing this game formulation requires high computational complexity in time and space. The double oracle, a constraint generation method, is deployed to overcome this bottleneck.

### 3.5 MODEL LEARNING PARAMETERS

Optimizing the model parameters $\boldsymbol{\theta}^i$ and $\boldsymbol{\theta}^{i,j}$ is another major step in Min-Max ML-ZSL. We employ standard tools from convex optimization like AdaGrad Duchi et al. (2011). The gradients $\mathbf{g}^i$ and $\mathbf{g}^{i,j}$ for a single sample $(\mathbf{x}, \mathbf{y})$ are as follows:

$$\mathbf{g}_t^i = \left\{ \mathbb{E}_{\mathbf{Y}_{maxi}|\mathbf{x} \sim P_{maxi}}[\boldsymbol{\phi}_t^i(Y_{maxi}{}^i, \mathbf{x})] - \boldsymbol{\phi}_t^i(y^i, \mathbf{x}) \right\}, \mathbf{g}_{joint}^i = \left\{ \mathbb{E}_{\mathbf{Y}_{maxi}|\mathbf{x} \sim P_{maxi}}[\boldsymbol{\phi}_{joint}^i(Y_{maxi}{}^i, \mathbf{x})] - \boldsymbol{\phi}_{joint}^i(y^i, \mathbf{x}) \right\}$$

$$\mathbf{g}_{lw}^i = \left\{ \mathbb{E}_{\mathbf{Y}_{maxi}|\mathbf{x} \sim P_{maxi}}[\boldsymbol{\phi}_{lw}^i(Y_{maxi}{}^i, \mathbf{x})] - \boldsymbol{\phi}_{lw}^i(y^i, \mathbf{x}) \right\},$$

$$\mathbf{g}_{pw}^{i,j} = \left\{ \mathbb{E}_{\mathbf{Y}_{maxi}|\mathbf{x} \sim P_{maxi}}[\boldsymbol{\phi}_{pw}^{i,j}(Y_{maxi}{}^i, Y_{maxi}{}^j, \mathbf{x})] - \boldsymbol{\phi}_{pw}^{i,j}(y^i, y^j, \mathbf{x}) \right\}.$$  (9)

### 3.6 INFERENCE

Min-Max ML-ZSL model parameters are learned using the double oracle method McMahan et al. (2003) and AdaGrad Duchi et al. (2011) as optimization. In the inference step, the learned parameters are employed, and the algorithm is run. The probabilistic output of the min-max method as a distribution over possible graph-cuts (label vectors) for the maximizer is a possible support assignment set. The graph-cut (label vector) with the highest probability is chosen as the prediction.

The training phase is done using seen classes as training data. The testing data are from unseen and seen classes in the testing phase. Hence, nodes (seen/unseen classes) connected to $\mathbf{0}$ have the label $0$ meaning there is no entity of the classes in the sample. The other nodes connected to $\mathbf{1}$ have the label $1$ (i.e., there exist entities of these classes in the sample data) in our prediction.

In this model, nodes and edges play an essential role in transferring knowledge from seen classes to unseen classes. We stimulate the similarities among seen and unseen classes using graph edges. When two nodes (seen and unseen) are similar, they would have similar/close weights and features values for their outgoing edges to other nodes. Therefore, the edges and nodes of unseen classes' weights/values in the graph would have values to their similar classes in the seen set classes. Hence,

the learning parameter and the inference process are directed to make the correct prediction (min-cut in our method). For instance, an unseen class "fish" has close feature values to the feature values of the seen class "dolphin" on the edges and the graph structure representation (figure 1). It directs the min-cut in the right direction, recognizing an unseen class "fish" though the classifier has not been trained for this class. We employ this key idea and leverage comprehensive feature representations. **Overview of the Min-Max ZSL method:** Since our min-max ZSL method and feature representation have been explained in previous sections, we present it through a simple example for better illustration. Assume training a classifier on seen classes like lion, sheep, dog, dolphin, ocean, cow, and panda presented as graph nodes.

The classifier is trained over the training samples (seen classes) to make the right min-cut to connect present labels to node 1 and absent labels to node 0 in a multi-label problem. The test sample includes classes like "fish" and "sea". Therefore, the classifier is tested on a testing sample, including classes on which the classifier has not been trained.

Our proposed method and feature representation help the classifier to 1) learn the co-occurrence of classes like "dolphin" and "ocean" in the training phase. 2) recognizes the high hierarchy feature representation overlap of classes "fish" and "dolphin". 3) recognizes the high hierarchy feature representation overlap of classes "sea" and "ocean". 4) conclude the co-occurrence of classes "fish" and "sea" based on the edges and nodes values ($\theta.\phi(x)$) and make a right min-cut in the graph where connects classes "fish" and "sea" to the node "1" in the same side of the min-cut. In general, the co-occurrences, similarities, and hierarchy structure in our graph representation lead the classifier to behave the same on similar seen and unseen classes. Modeling a multi-label problem as a graph allows the classifier to learn the structure/relationship among the labels. This learned structure/relationship knowledge exists among similar classes and is transferred in the ZSL setting.

## 4 EXPERIMENTS

Our experiments include two main analyses: performance and convergence to flat/sharp minima. We first present our experimental studies on ML-ZSL methods' performance on small and large datasets. Next, we explain convergence to flat/sharp minima as an important metric to evaluate the generalization of ZSL methods. We run an analysis and show that the methods with convergence to flat minima result in better performance in the ZSL setting.

**Performance Analysis:** We benchmark our method on different datasets and compare them with state-of-the-art methods. Finally, we present more experiments and explain our baselines in Appendix, section "More Experiments".

In the first experiment, we consider two standard multi-label datasets: the PASCAL VOC2007 Everingham et al. (a) and VOC2012 Everingham et al. (b) datasets. The VOC2007 dataset includes 20 visual object classes. There exist 9963 images in total in this dataset, 5011 for training and 4952 for testing. The VOC2012 dataset contains 5717 and 5823 images for training and validation from 20 classes. We leverage the validation set for test evaluation. For each image, we extract 4096-dim visual features VGG19 Simonyan & Zisserman (2014) pre-trained on ImageNet. We split the seen classes into two disjoint subsets with an equal number of classes for training and validation to determine the hyper-parameters. In this setting, we train the model on the training set and choose hyper-parameters based on the test performance on the validation set. After parameter selection, the training and validation data are put back together to train the model for the final evaluation of unseen test data.

For our method, Min-Max, we employ the feature representations of CNN 6, 33 layers to present layer-wise and pairwise feature representation as it provides better feature representation for knowledge transfer (based on our experiments in Appendix, section "More Experiments"). The taxonomy features representation is computed using VOC 2006 taxonomy structure, and the seen-unseen standard split in Norouzi et al. (2013). We report our experimental results in table 1 using three different multi-label evaluation metrics: MiAP, micro-F1 (presented as mi-F1 in table 1), and macro-F1 (presented as ma-F1 in table 1). The Mean image Average Precision (MiAP) Li et al. (2016) evaluates how well the labels are ranked on a given image based on the prediction scores. Micro-F1 and macro-F1 measure how well the predicted labels match the test data's ground truth labels.

**Discussion:** In table 1, we present the experiment results of six ZSL methods (Conse Norouzi et al. (2013), LatEn-M Xian et al. (2016), DMP Fu et al. (2015), Fast0Tag Zhang et al. (2016), and TAEP-C

| Methods | VOC2007 | | | VOC2012 | | |
|---|---|---|---|---|---|---|
| | MiAP | mi-F1 | ma-F1 | MiAP | mi-F1 | ma-F1 |
| ConSE | 45.65 | 27.64 | 24.83 | 45.34 | 30.83 | 26.11 |
| LatEm-M | 47.37 | 28.39 | 27.16 | 46. 17 | 27.46 | 26.37 |
| DMP | 53.23 | 35.53 | 38.82 | 52.32 | 34.31 | 35.94 |
| Fast0Tag | 52.45 | 34.67 | 35.91 | 51.09 | 33.13 | 33.97 |
| TAEP-C | 54.64 | 36.56 | 37.87 | 52.84 | 33.26 | 34.23 |
| GEN | 54.92 | 36.73 | 37.65 | 53.89 | 36.24 | 36.10 |
| BiAM | 55.28 | 36.83 | 37.58 | 54.89 | 37.63 | 38.65 |
| Min-Max | 58.11 | 38.86 | 39.08 | 57.10 | 39.22 | 42.85 |

Table 1: Average comparison results (%) over ten runs on zero-shot multi-label image classification.

| Method | Task | NUS-WIDE ( #seen / #unseen = 925/81) | | | | | | | Open-Images ( #seen / #unseen = 7186/400) | | | | | | |
|---|---|---|---|---|---|---|---|---|---|---|---|---|---|---|---|
| | | K = 3 | | | K = 5 | | | mAP | K = 10 | | | K = 20 | | | mAP |
| | | P | R | F1 | P | R | F1 | | P | R | F1 | P | R | F1 | |
| CONSE | ZSL | 17.5 | 28.0 | 21.6 | 13.9 | 37.0 | 20.2 | 9.4 | 0.2 | 7.3 | 0.4 | 0.2 | 11.3 | 0.3 | 40.4 |
| | GZSL | 11.5 | 5.1 | 7.0 | 9.6 | 7.1 | 8.1 | 2.1 | 2.4 | 2.8 | 2.6 | 1.7 | 3.9 | 2.4 | 43.5 |
| LabelEM | ZSL | 15.6 | 25.0 | 19.2 | 13.4 | 35.7 | 19.5 | 7.1 | 0.2 | 8.7 | 0.5 | 0.2 | 15.8 | 0.4 | 40.5 |
| | GZSL | 15.5 | 6.8 | 9.5 | 13.4 | 9.8 | 11.3 | 2.2 | 4.8 | 5.6 | 5.2 | 3.7 | 8.5 | 5.1 | 45.2 |
| Fast0Tag | ZSL | 22.6 | 36.2 | 27.8 | 18.2 | 48.4 | 26.4 | 15.1 | 0.3 | 12.6 | 0.7 | 0.3 | 21.3 | 0.6 | 41.2 |
| | GZSL | 18.8 | 8.3 | 11.5 | 15.9 | 11.7 | 13.5 | 3.7 | 14.8 | 17.3 | 16.0 | 9.3 | 21.5 | 12.9 | 45.2 |
| One Attention | ZSL | 20.0 | 31.9 | 24.6 | 15.7 | 41.9 | 22.9 | 12.9 | 0.6 | 22.9 | 1.2 | 0.4 | 32.4 | 0.9 | 40.7 |
| | GZSL | 10.4 | 4.6 | 6.4 | 9.1 | 6.7 | 7.7 | 2.6 | 15.7 | 18.3 | 16.9 | 9.6 | 22.4 | 13.5 | 44.9 |
| LESA (M=10) | ZSL | 25.7 | 41.1 | 31.6 | 19.7 | 52.5 | 28.7 | 19.4 | 0.7 | 25.6 | 1.4 | 0.5 | 37.4 | 1.0 | 41.7 |
| | GZSL | 23.6 | 10.4 | 14.4 | 19.8 | 14.6 | 16.8 | 5.6 | 16.2 | 18.9 | 17.4 | 10.2 | 23.9 | 14.3 | 45.4 |
| GEN | ZSL | 26.6 | 42.8 | 32.8 | 20.1 | 53.6 | 29.3 | 25.7 | 1.3 | 42.4 | 2.5 | 1.1 | 52.1 | 2.2 | 43.0 |
| | GZSL | 30.9 | 13.6 | 18.9 | 26.0 | 19.1 | 22.0 | 8.9 | 33.6 | 38.9 | 36.1 | 22.8 | 52.8 | 31.9 | 49.7 |
| DRM | ZSL | 26.4 | 42.4 | 32.5 | 19.8 | 53.1 | 28.8 | 25.3 | 10.3 | 42.4 | 16.6 | 12.9 | 52.1 | 20.7 | 43.7 |
| | GZSL | 30.3 | 13.1 | 18.3 | 25.7 | 22.5 | 24.0 | 7.8 | 33.9 | 39.1 | 36.3 | 24.2 | 50.6 | 32.7 | 49.9 |
| BiAM | ZSL | 28.7 | 39.8 | 33.4 | 21.6 | 53.4 | 30.7 | 26.0 | 4.6 | 44.1 | 8.3 | 3.8 | 9.5 | 5.4 | 45.3 |
| | GZSL | 30.4 | 14.0 | 19.2 | 28.5 | 18.6 | 22.1 | 9.2 | 36.9 | 35.5 | 36.2 | 24.5 | 51.6 | 33.3 | 49.9 |
| SKG | ZSL | **29.4** | 31.7 | 30.5 | 20.2 | 53.0 | 29.25 | 26.4 | 34.1 | 33.7 | 33.9 | 23.1 | 44.38 | 30.5 | 47.3 |
| | GZSL | 23.7 | 11.5 | 15.5 | 22.8 | 25.9 | 24.3 | 6.4 | 18.5 | 31.6 | 23.3 | 22.1 | 36.9 | 27.6 | 48.6 |
| **Min-Max** | ZSL | **29.3** | **44.7** | **35.4** | **22.7** | **57.4** | **31.8** | **30.9** | 5.3 | 44.5 | 9.5 | 4.8 | 54.4 | 8.8 | **48.3** |
| | GZSL | **33.0** | **24.5** | **28.1** | **32.3** | **30.5** | **31.4** | **11.0** | **37.9** | **41.8** | **39.8** | **27.6** | **55.3** | **36.8** | **51.5** |

Table 2: State-of-the-art comparison for ZSL and GZSL tasks on the NUS-WIDE and Open Images datasets. We report the results in terms of mAP and F1 score at $K \in \{3, 5\}$ for NUS-WIDE and $K \in \{10, 20\}$ for Open Images. The best results are in bold.

Ye & Guo (2019)) on VOC2007 and VOC2012 datasets. As it is shown, the Min-Max provides the maximum MiAP compared with other methods on both datasets. Considering MiAP, Min-Max outperforms Conse and BiAm by 12.4% and 2.8% on the VOC2007 dataset. The experiment results on VOC2012 present the strong performance of Min-Max compared to other methods.

Table 2 shows the state-of-the-art comparison for ZSL and generalized ZSL (GZSL) ML classification on NUS-WIDE and Open-Images datasets. The results are reported in terms of mAP and F1 score at *top-K* predictions ($K \in \{3, 5\}$). **NUS-WIDE:** For the ZSL task, SKG Lee et al. (2018) and BiAM approaches Narayan et al. (2021) achieve improved performance over GEN Gupta et al. (2021), LESA Huynh & Elhamifar (2020) , DRM Ji et al. (2020) and Fast0Tag, with mAP scores of 26.4 and 26.0. Our approach (Min-Max) achieves state-of-the-art results with an absolute gain of 4.5% in mAP over the best existing approach, SKG. Similarly, our approach obtains consistent improvement in classification performance over the state-of-the-art in terms of F1 score ($K \in \{3, 5\}$).

For the GZSL task, the min-max approach achieves a mAP score of 11.0, outperforming BiAM, GEN, and DRM with absolute gains of 1.8%, 2.1%, and 3.2% respectively. Similarly, our approach achieves consistent improvement in classification performance with absolute gains of 8.9% and 9.3% over BiAM in terms of F1 score at $K=3$ and $K=5$, respectively.

**Open Images:** The results are reported in terms of mAP and F1 score at *top-K* predictions ($K \in \{10, 20\}$). Compared to the NUS-WIDE dataset, Open Images has more labels. It makes the ranking problem within an image more challenging, reflected by the lower F1 scores in the table. For the ZSL task, the min-max method performs favorably against other baselines with F1 scores of 9.5 and 8.8 at $K=10$ and $K=20$, respectively. It is worth noting that this dataset has 400 unseen labels resulting in a challenging ZSL problem. Our Min-max approach outperforms the state-of-the-art, in terms of both F1 and mAP, for the GZSL task. In Narayan et al. (2021), the mAP on OpenImage dataset is reported as 73.6 and 84.5 for ZSL and GZSL. We ran these experiments leveraging the code provided by the authors but could not replicate the reported mAP results.

## 4.1 Sharp and Flat Minima

Hochreiter Hochreiter & Schmidhuber (1995) launched the discussion about flat and sharp local minima in deep neural networks. A flat minimizer (learning parameter) $\bar{\theta}$ is the one for that the

function varies slowly in a relatively large neighborhood of $\bar{\theta}$ leading to low sensitivity. In a sharp minimizer $\hat{\theta}$, the function decreases rapidly in a small neighborhood of $\hat{\theta}$ resulting in high sensitivity. The low sensitivity of the training function at a flat minimizer positively impacts the ability of the trained model to generalize on new data; Rissanen (1983) presents that statistical models that require fewer bits to describe are of low complexity and generalize better Keskar et al. (2016). Keskar et al. (2016) shows that learning systems that find sharp local minim lead to poor generalization.

**Sharpness Metric:** The sharpness of an optimizer/ method can be measured by the magnitude of the eigenvalues of $\nabla^2 f(x)$. This computation requires high computational costs in deep neural networks. We use the sensitivity measure Keskar et al. (2016) to avoid the infeasible high computational cost.

The sensitivity measure (sharpness measure) is the largest value that the function $f$ can achieve in a small neighborhood of the local minima of the function $f$. To make an accurate maximization process, the maximization process is done both in the entire space $R^n$ (where $n$ is the feature space dimension) and in random manifolds. For this purpose, the matrix $S_{n*m}$, where $m$ represents the manifold dimension, with random values is employed. We set $m = 100$ in our experiments. The constraint set $\mathcal{C}_\epsilon$ is defined to ensure invariance of sharpness to problem dimension and sparsity.

$$\mathcal{C}_\epsilon = \{z \in \mathbb{R}^m : -\epsilon(|(S^+x)_i| + 1) \leq z_i \leq \epsilon(|(S^+x)_i| + 1), \forall i \in \{1, 2, \cdots, m\}\}. \tag{10}$$

Here $S^+$ denotes the pseudo-inverse of $S$; hence, $\epsilon$ controls the box size. Given $x \in \mathbb{R}^n$, $\epsilon > 0$ and $S \in \mathbb{R}^{n \times m}$, the measure of sharpness (or sensitivity) of function $f$ at $x$ is computed as following:

$$\phi_{x,f}(\epsilon, S) := \frac{(\max_{y \in \mathcal{C}_c} f(x + Sy)) - f(x)}{1 + f(x)} \times 100. \tag{11}$$

The sharp minima have a different definition in convex optimization literature Ferris (1988). To better illustrate curvature changes in a ZSL setting, we run an experiment on different datasets and calculate the sharpness value for consecutive tasks following Keskar et al. (2016) experiment setting. We report the result of experiments in table 3. In this experiment setting, we explore the full-space $(S = I_n)$ setting the value of $\epsilon$ to $(10^{-3})$. The higher the value of sharpness presents the more method sensitivity. We report more experiments considering $\epsilon$ as $(5.10^{-4})$ in the appendix.

| Method/Datasets | VOC2007 | VOC 2012 | NUS-1000 |
|---|---|---|---|
| $\epsilon = 10^{-3}$ | | | |
| ConSE | $87.12 \pm 7.87$ | $93.37 \pm 5.61$ | $98.63 \pm 4.52$ |
| LatEm | - | $67.21 \pm 4.32$ | $97.83 \pm 5.38$ |
| Fast0Tag | $53.86 \pm 5.53$ | $43.65 \pm 6.34$ | $90.59 \pm 7.26$ |
| DMP | $51.96 \pm 4.90$ | $40.45 \pm 6.54$ | $80.73 \pm 4.62$ |
| TAEP-C | $49.87 \pm 3.66$ | $39.79 \pm 9.51$ | $88.94 \pm 8.49$ |
| LESA | $45.31 \pm 2.23$ | $37.51 \pm 3.48$ | $83.24 \pm 5.49$ |
| GEN | $41.34 \pm 4.49$ | $34.78 \pm 3.84$ | $80.27 \pm 5.43$ |
| BiAM | $40.52 \pm 4.64$ | $33.69 \pm 4.21$ | $79.15 \pm 4.63$ |
| Min-Max | $35.7 \pm 3.94$ | $31.24 \pm 5.20$ | $75.67 \pm 6.78$ |

Table 3: Sharpness value considering $\epsilon = 10^{-3}$.

**Discussion** As it is presented in table 3, Min-Max demonstrates much less sharpness compared with the other methods on three datasets. It presents the effectiveness of different approaches in leading the model to flat minima in the optimization process. The sharp minima in a multi-label setting happen when the loss function increases rapidly only along a small dimensional subspace. The function is almost flat in other directions. The Min-Max method smooths this significant gradient update. This experiment demonstrates the importance of convergence to flat minima in the optimization process for ZSL frameworks. As presented in table 3, the methods that provide better performance in the ZSL setting have less sensitivity (sharpness).

## 5   Conclusion

This paper proposes a novel probabilistic framework for ZSL, including extensive feature representation and probabilistic cuts for ZSL. Our method relies on probabilistic graph cut output, effectively capturing the implicit features and finding the most important and closest attributes to unseen class attributes at testing time.

The Min-Max method significantly outperforms the existing ZSL state-of-the-art methods on well-known datasets. Besides, we run a sharpness analysis considering different ZSL methods. Our study suggests the importance of low sharpness value in better generalizing multi-label ZSL methods.

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
