# OpenReview forum: "Min-Max Zero-Shot Multi-Label Classification"
_ICLR.cc/2023/Conference — Submitted to ICLR 2023_

### Official Review · Reviewer_TsJ7 · 2022-10-22

**Confidence:** 3
**Correctness:** 2
**Technical Novelty And Significance:** 2
**Empirical Novelty And Significance:** 2
**Recommendation:** 3

**Clarity, Quality, Novelty And Reproducibility:**

The proposed method is somewhat novel with a min-max graph cut strategy to tackle multi-label ZSL. However, this work cannot claim the first work that provides a framework using structured prediction to model multi-label ZSL. The quality of the paper is far from ready for publication. As technical writing is not very clear, it is also not clear if the proposed method and the results are reproducible.

**Strength And Weaknesses:**


Strengths:
- The proposed method is somewhat novel with a min-max graph cut strategy to tackle multi-label ZSL.
- There are some rigorous experiments to show the effectiveness of the method and the generalization capability using the sharpness metric.
- The performance is superior compared to some selected state-of-the-art methods for all experiments.

Weaknesses:
- The structured prediction for multi-label zero shot learning is not first proposed in this work, thus cannot be included as a contribution. Previous works by Lee et al. (“Multi-Label Zero-Shot Learning with Structured Knowledge Graphs”) and Huang et al., (“Multi-label Zero-shot Classification by Learning to Transfer from External Knowledge”) have proposed similar strategies using rstructured graph networks and relational graph convolutional networks, respectively.
- This work has an Inconsistent term whether the problem is zero-shot multi-label classification or multilabel zero-shot learning.
- As understood from the description in this work, there would be a drawback in representing the labels with low occurance and also create “blind” correlation without any causation between attributes and classes.
- The notation and mathematical expressions of this paper are inconsistent, unstructured, and difficult to follow. Sometimes a label vector is denoted as $\bold{y}$ and $\bold{Y}$. Another thing is y^{entity} is written as y_{entity}. Variable x is sometimes bold.
- The proposed method is not clear at all, the readability for this work is very low. After several times reading the mathematical expressions in Section 3, the proposed method is unclear. For instance, why does Eq. 3 uses abs(diff. between two embeddings from word vector) as the divisor? What are parameters to be optimized in Eq. 8 ($\Theta^{i,j}$ is not clear)?
- $P_{data}$, $P_{maxi}$ and $P_{mini}$ are not elaborated further in text.
- Some abbreviations for comparing with all SOTA methods in all tables are not clear and referred in text (e.g., One Attention, LabelEM, and no explanation of M=10 for LESA). Please add complete references in each table.
- There is no direct comparison with the methods with similar approaches e.g., Graph convolutional networks.
- The “Discussion” paragraph in Page 7 has no discussion at all. There are only some references and the report of the results.
- Some grammatical errors: by Mulan dataset, features values, 1) learn 2) recognizes 3) recognizes. Please review the manuscript in more details as there are notable mistakes in writing.


**Summary Of The Paper:**


This paper presents a novel method for zero-shot multi-label classification where the model is trained on seen classes and tested on unseen classes without further model tuning. The strategy uses min-max zer-sum game between the maximizer and the minimizer with minimum cost graph cuts. The proposed method has four additional losses besides the Hamming loss applied for each element of a target vector, namely taxonomy, joint, layer-wise, and pairwise losses.  The proposed method is proven superior compared to the state-of-the-art in multi-label ZSL on NUS-WIDE and Open Images.


**Summary Of The Review:**

This work has some potentials to tackle the problem of multi-label zero-shot learning looking at excellent performance compared to SOTA. However, the presentation is lacking of clarity, and there are also some claims that need to be tone-down to make the contributions clear.

---

### Official Review · Reviewer_WnWm · 2022-10-25

**Confidence:** 3
**Correctness:** 4
**Technical Novelty And Significance:** 3
**Empirical Novelty And Significance:** 3
**Recommendation:** 6

**Clarity, Quality, Novelty And Reproducibility:**

Clarity: As I said above, the paper can be better organized. But the authors tried to use examples to walk through the proposed method, which should be encouraged.

Quality & Novelty: The paper seems novel to me as it does not resemble the prior arts. Considering its compelling empirical results, the overall quality of the paper is good.

Reproducibility: The authors provided the algorithm in the appendix. But it lacks enough training and implementation details to help readers reproduce the method.

**Strength And Weaknesses:**

Pros:
1. The proposed method seems novel in the respective area
2. The method achieves state-of-the-art performance on various benchmarks for ML-ZSL. It also performs very competitive results on conventional ZSL/GZSL.

Cons:
1. The organization of the paper can be improved. I find myself having to go back and forth to understand certain parts when reading. The presentation of the formulae can also be improved to make them more straightforward.

**Summary Of The Paper:**

This paper proposes a new method, Min-Max, for multi-label zero-shot learning (ML-ZSL). The proposed method is evaluated on multiple benchmarks, including ZSL, GZSL, and multi-label ZSL datasets, and shows compelling results.

**Summary Of The Review:**

Summary: Given this paper's novelty and quality, I am inclined to accept it. However, I am not very familiar with this area, and I might be wrong regarding the novelty.

---

### Official Review · Reviewer_ZVzZ · 2022-11-03

**Confidence:** 3
**Correctness:** 3
**Technical Novelty And Significance:** 3
**Empirical Novelty And Significance:** 3
**Recommendation:** 5

**Clarity, Quality, Novelty And Reproducibility:**

Clarity: the clarity can be improved, including figures, descriptions, and logical flow.
Quality: the quality is moderate.
Novelty: there are novelties in the proposed method but not that significant.
Reproducibility: the authors are expected to release the code to the public.

**Strength And Weaknesses:**

Strength:
+ The explored task is highly challenging and beneficial to practical applications.
+ Constructing class relations based on the graph structure is interesting.
+ Empirical results seem good compared to the presented baselines.

Weaknesses or questions:
- All these feature representation manners are simply combined without in-depth analysis. Can you provide any insights on them? In addition, how to accurately construct the relationship between seen classes and unseen classes based on these features?
- More related studies are welcomed. For example, visual-language models have recently shown highly promising performance for zero-shot learning. It would be better to discuss them.
- Figure 4 is unclear. How to learn the classifier for making min-cut predictions for unseen classes during training?
- It is better to give some visualization results for some images and their prediction probabilities for different unseen classes.
 In addition, please also provide some failure cases to discuss the limitations.
- In Section 4.1, it is better to visualize the loss surface to see whether it is flatter than other baselines.

Minor issues:
- The format of in-text citations is not good and can be improved.
- All figures are not formal and academic. The authors are suggested to make them more formal.

**Summary Of The Paper:**

This paper explores a challenging task of zero-shot multi-label classification. To address this task, the authors propose to combine multiple types of feature representations to represent data, including Word-Net hierarchy, word2vec features, CNN layer-wise features and co-occurrence statistics. Based on these features, a new min-max zero-shot multi-label learning method is proposed to learn classifiers to predict test data for unseen classes. Experimental results show the effectiveness of the proposed method.

**Summary Of The Review:**

Overall, although there are merits in this paper, some issues affect its clarity and contributions. It is better for authors to address the concerns for further improving the quality of the paper.

---

### Decision · Program_Chairs · 2023-01-20

**Decision:**

Reject

**Justification For Why Not Higher Score:**

This paper got mixed ratings, 2 for reject and 1 for borderline accept. But the reviewer giving borderline accept does not give strong evidence to support accept. The reviewer also admits he/she is not familiar with this field. The reviewers giving reject share similar concerns on the technical depth, method novelty and presentation quality. The authors do not respond to the first reviewer, leaving the questions unanswered. The authors provide clarification on the technical details as requested by the third reviewer. But their response does not address the fundamental concerns on the technical novelty and depth.

**Justification For Why Not Lower Score:**

N/A

**Metareview: Summary, Strengths And Weaknesses:**

This paper explores zero-shot multi-class classification problem. To tackle this challenging problem, it proposes a new probabilistic models to incorporate multiple different features and establish between-sample relationship for classification.

Strength

- It explores a challenging problem.

Weakness

- The proposed approach is straightforward. It combines multiple types of features to help classification. But it lacks insights on the feature selection.

- The proposed approach is also similar to existing works,SKG (structured Knowledge graph by Lee et al 2018), making its novelty limited.

- The presentation quality is not good. The notations are not clear. Some technical details are lacking.